



# Response of atmospheric composition to COVID-19 lockdown measures during Spring in the Paris region (France)

Jean-Eudes Petit[1], Jean-Charles Dupont[2], Olivier Favez[3], Valérie Gros[1], Yunjiang Zhang[1,3], Jean Sciare[1*], Leila Simon[1,3], François Truong[1], Nicolas Bonnaire[1], Tanguy Amodeo[3], Robert Vautard[1], Martial Haeffelin[4]

[1]Laboratoire des Sciences du Climat et de l'Environnement, CEA/Orme des Merisiers, Gif-sur-Yvette, France
[2]Institut Pierre Simon Laplace, Ecole Polytechnique, UVSQ, Université Paris-Saclay, Palaiseau, France
[3]Institut National de l'Environnement Industriel et des Risques, Parc Technologique ALATA, Verneuil en Halatte, France
[4]Institut Pierre Simon Laplace, Ecole Polytechnique, CNRS, Université Paris-Saclay, Palaiseau, France
* Now at Cyprus Institute, Nicosia, Cyprus

*Correspondence to*: Jean-Eudes Petit (jean-eudes.petit@lsce.ipsl.fr)

**Abstract.** Since early 2020, the COVID-19 pandemic has led to lockdowns at national scales. These lockdowns resulted in large cuts of atmospheric pollutant emissions, notably related to the vehicular traffic source where daily commuting of light-duty vehicles was almost completely stopped in numerous urban areas worldwide, especially during Spring 2020. As a result, air quality changed in manners that are still currently
under investigation. Long-term in-situ monitoring of atmospheric composition provides, to this perspective, essential information. However, a robust quantitative assessment of the impact of lockdown measures on ambient concentrations is hindered by weather variability. Basic comparisons with previous years may thus be flawed, especially regarding secondary pollutants, whose concentrations strongly depends on meteorological conditions. In order to circumvent this difficulty, an innovative methodology has been
developed. The Analog Application for Air Quality (A³Q) method is based on the comparison of each day of lockdown to a group of analog days having similar meteorological conditions. The A³Q method has been successfully evaluated and applied to a comprehensive in-situ dataset of primary and secondary pollutants obtained at the SIRTA observatory, a suburban background site of the Paris megacity (France). The overall slight decrease of $PM_1$ concentrations (-14%) compared to business-as-usual conditions conceals contrasting
behaviours. Primary traffic tracers ($NO_x$ and traffic-related carbonaceous aerosols) dropped by 42-66% during the lockdown period. Further, the A³Q method enabled us to characterize of changes triggered by $NO_x$ decreases. Particulate nitrate and secondary organic aerosols (SOA), two of the main springtime aerosol components in North-Western Europe, decreased by -45% and -25%, respectively. A $NO_x$-relationship emphasizes the interest of $NO_x$ mitigation policies at the regional (i.e. city) scale, although long-range
pollution advection sporadically overcompensated regional decreases. Variations of the oxidation state of SOA suggests discrepancies in SOA formation processes. At the same time, the expected ozone increase (+20%) underlines the negative feedback of NO titration. These results provide a quasi-comprehensive observation-based insight on mitigation policies regarding air quality in future low-carbon urban areas.

## 1. Introduction

With the worldwide spreading of the SARS-COV-2 coronavirus, the COVID-19 outbreak has been responsible of millions of premature deaths. In order to slow down contagion rates, social interactions have progressively been limited until the establishment of strict lockdowns at national scales (Anderson et al., 2020) enforced





during several weeks, especially during Spring 2020 in Europe. The corresponding stay-at-home orders resulted in a sudden halt of economic activities, and, as a consequence, in an unprecedented drop of emission

of pollution sources. To this perspective, and despite tragic death records, these lockdowns are unique opportunities to characterize an extreme end of mitigation policy scenarios, and future low-carbon megacities from direct observations.

Scientific initiatives are thriving across the globe in order to assess the impact of lockdowns on air quality. They report, for most, a sharp decrease of nitrogen oxides ($NO_x$) concentrations, as well as an increase of

trosopheric ozone (e.g. China: Le et al. (2020); India: Mahato et al. (2020); USA: Liu et al. (2020a); Europe: Sicard et al. (2020); Grange et al. (2020); South-America: Siciliano et al. (2020)) as a response to stay-at-home orders. However, despite this luxuriance of literature, the assessment of air quality implications of large cuts in urban pollutant emissions is strongly hampered by meteorological variability, which is one of the main drivers of air pollution temporality. Without climatologically representative values, comparisons of

concentrations observed during and outside the lockdown periods shall thus free themselves from differences in weather. The robustness of this assessment depends on the way meteorology is handled and on what "reference period" is chosen to compare with the "lockdown period". Recently, advances on machine-learning (ML) approaches have enabled to unknot the contributions of meteorological conditions into weather-normalized timeseries (e.g. Stirnberg et al., 2021). ML has successfully been applied mainly on

$NO_x$ and $O_3$ in various urban areas (Petetin et al., 2020; Grange et al., 2020), but little information are being gathered on PM and its chemical constituents. Air quality shall not be restrained to $NO_x$, $O_3$ and $PM_x$ only, and limited number of studies so far have treated air quality as a whole, notably by taking PM chemistry into account (Kroll et al., 2020). Indeed, PM is composed of several different fractions, from organic to inorganic, and from primary to secondary pollutants, with diverse sources and transformation processes. Any

concentration change of PM may derive from various compensatory feedbacks which are not characterized, limiting therefore our understanding of the impacts of lockdown on air quality. Moreover, Springtime in North-Western Europe is usually associated with high PM pollution episodes dominated by secondary material (mainly ammonium nitrate and sulfate, and secondary organic aerosols) as shown in Petit et al. (2015), Beekmann et al. (2015), Petit et al. (2017a) for instance. To this perspective, the complex chemistry of PM must

be taken into account, emphasizing the need of in-situ aerosol chemical characterization data.

The present study aims at reconciling a robust and innovative methodology with a quasi-comprehensive in-situ dataset, acquired within the Paris region (France). The 12-million inhabitants of the region, representing around 20% of the total French population, were placed under lockdown from March 17th, 2020 to May 10th, 2020, further designated as LP2020.


## 2. In-situ characterization of the atmospheric composition

### 2.1. Instrumentation

In-situ measurement datasets used in this study have been primarily obtained at the SIRTA atmospheric observatory (2.15°E, 48.71°N; Haeffelin et al., 2005), a facility which contributes to the EU-research





infrastructure ACTRIS (https://www.actris.eu), following its Quality assurance / Quality control guidelines. The chemical composition of major submicron non-refractory species has been monitored since the end of 2011 using an Aerosol Chemical Speciation Monitor (ACSM, Ng et al., 2011), constituting the longest ACSM dataset worldwide. Measurement principles of the ACSM are extensively described elsewhere (Budisulistiorini et al., 2014; Zhang et al., 2019; Poulain et al., 2020). Here, 30-min concentrations of submicron

Organic Aerosols (OA), Nitrate ($NO_3$), Sulfate ($SO_4$), Ammonium ($NH_4$) and Chloride (Cl) were corrected from collection efficiency (Middlebrook et al., 2012) (CE). The ACSM at SIRTA was regularly calibrated using 300-nm ammonium nitrate and ammonium sulfate particles to derive Ionization Efficiencies (IE), and successfully participated in ACTRIS intercomparaison exercises (Crenn et al., 2015; Freney et al., 2019).

    The black carbon dataset consists in Aethalometer measurements (Drinovec et al., 2015). It is composed of

subsequent and harmonized datasets obtained from AE31 (01/2011-03/2013) and AE33 , 03/2013-06/2020) devices and applying a common validation procedure (Petit et al., 2017b). Briefly, for each elementary measurement data point (5-min and 1-min time-base for AE31 and AE33, respectively), $BC_\lambda$ concentrations were set as invalid when below -LoD (Limit of Detection, 100 ng/m³); for $BC_{950nm} \geq 200$ ng/m³, the spectral dependence was calculated from the linear regression of $\ln(\lambda)$ versus $\ln(B_{atn})$. Measurements were considered

valid for a r² (of this linear regression) higher than 0.9 and Aerosol Angström Exponent (AAE) comprised between 0.8 and 3.

    Daily concentrations of nitrogen monoxide (NO) and nitrogen dioxide ($NO_2$) were retrieved from 1-min measurements performed with a T200UP Teledyne instrument, equipped with a blue light photolytic converter and a Nafion dryer. The instrument has been regularly calibrated with a reference standard from

National Physics Laboratory (Teddington, UK) and NO and $NO_2$ concentrations have been corrected from ozone interference. The $NO_x$ analyzer has participated to the 2 ACTRIS intercomparaison excercises organized at Hohenpeissenberg in 2012 and 2016 and has shown a good comparability with other instruments. Other $NO_x$ observations throughout the Paris region between 2012 and 2020 were retrieved from the regional air quality monitoring structure (Airparif, https://www.airparif.asso.fr). To this respect, urban $NO_x$

concentrations refer here to the average of all urban background stations measuring $NO_x$.

    As the ozone instrument from SIRTA experienced a major breakdown in 2020, daily ozone ($O_3$) concentrations were obtained between 2012 and 2020 from a peri-urban station in Les Ulis (2.165°E, 48.68°N), operated by Airparif. This station is located around 10 km away from SIRTA and $O_3$ from Les Ulis has shown a good comparability with $O_3$ from SIRTA during the period of common measurements.

Meteorological variables at SIRTA (wind speed and direction, temperature, relative humidity and pressure) were provided from the ReObs database (Chiriaco et al., 2018).

## 2.2. Source apportionment of carbonaceous aerosols

    A source apportionment study of OA was carried out by Positive Matrix Factorisation (PMF, Paatero and

Tapper, 1994) from January to May 2020. The analysis has been carried out seasonally (January-February and March-April-May) in order to prevent from the seasonality of the profiles of secondary factors (Canonaco et al., 2015). Profiles of Hydrocarbon-like Organic Aerosols (HOA), Biomass Burning Organic Aerosols (BBOA)





were constrained with a random a-value approach (Canonaco et al., 2013), a third factor being left unconstrained (Oxygenated Organic Aerosol, OOA). The criteria approach of SoFi Pro (Canonaco et al., 2020)

was then used to select satisfactory solutions over 100 runs, from the R-Pearson correlation of HOA vs $BC_{ff}$, HOA vs $NO_x$ and BBOA vs $BC_{wb}$. Results obtained here enrich the existing timeseries (Zhang et al., 2019) from June 2011 to March 2018; together they form one of the longest ACSM PMF timeseries published to date. MO-OOA and LO-OOA were summed as OOA.

The oxidation properties of secondary organic aerosols were characterized by removing the contribution of

primary factors to the OA matrix, as follows:

$$f_i^{SOA} = \frac{m/z_i - (f_i^{HOA} \cdot [HOA] + f_i^{BBOA} \cdot [BBOA])}{[OA] - ([HOA] + [BBOA])}$$

From there, $O:C_{SOA}$, $H:C_{SOA}$ and $OSc_{SOA}$ were calculated from the equations provided in (Aiken et al., 2008; Kroll et al., 2011; Canagaratna et al., 2014). These equations provide only qualitative information for ACSM data, but are sufficient to characterize a change, since they are uniformly applied throughout the dataset.


Fossil-fuel ($BC_{ff}$) and Biomass burning ($BC_{wb}$) fractions were estimated from aethalometer measurements (Sandradewi et al., 2008). Since the choice of $\alpha_{ff}$ and $\alpha_{wb}$ is critical and given the size of our dataset, their determination was based on the statistical hourly distribution of the AAE (Fig. S1). The value of 1.85 was chosen for $\alpha_{wb}$, corresponding to a maximum frequency during the night. This value is close to the values used

previously at SIRTA (Zhang et al., 2019), and from the recommended value of 1.72 (Zotter et al., 2016). $PM_{ff}$ and $PM_{wb}$ were estimated from $BC_{ff}$ and $BC_{wb}$ concentrations, respectively, following the conversion factors of 2 and 10.3 found for SIRTA during the same season (Petit et al., 2014).

The full timeseries between 2012 and 2020 is presented in Figure S2.


**2.3. Backtrajectory calculation**

120-h backtrajectories ending at SIRTA (49.15°E, 2.19°N) at 500m a.g.l. were calculated every 6h from 2012 to 2020 with the PC-based version of HYSPLIT (Stein et al., 2015) using 1°x1° Global Data Assimilation System

(GDAS) files. Calculations were controlled by ZeFir (Petit et al., 2017a).

**3. Methodology to estimate the impact of lockdown measures : an attempt to compare apples to apples**

**3.1. Choice of the comparison reference period**

The assessment of lockdown impact on air quality lies on the use of a reference period, which is assumed to

be representative of business-as-usual conditions during LP2020, following:

$$\%change = 100 \cdot \frac{LP_{2020} - ref}{ref}$$





However, it can lead to some variability in the results, depending on the chosen reference period, representing therefore a critical issue. The major danger would be to apply methodologies unquestioningly,

without verifying the strong inherent hypothesis that data are comparable. For instance, one could consider the weeks preceding the lockdown period as a satisfactory reference (e.g., Toscano and Murena, 2020; Dantas et al., 2020; Otmani et al., 2020). In the case of SIRTA, we tried to compare the pollutant concentrations averaged over the lockdown period (LP 2020) to the ones calculated for the 20 previous days (pLP 2020 : 25/02/2020 - 16/03/2020). As shown in Figure 1, applying such a method led to the determination of

significant concentration increases for all pollutants during the lockdown period (e.g., + 83% in $NO_x$, +439% in $PM_1$), which seems to contradict the observed drop of traffic.

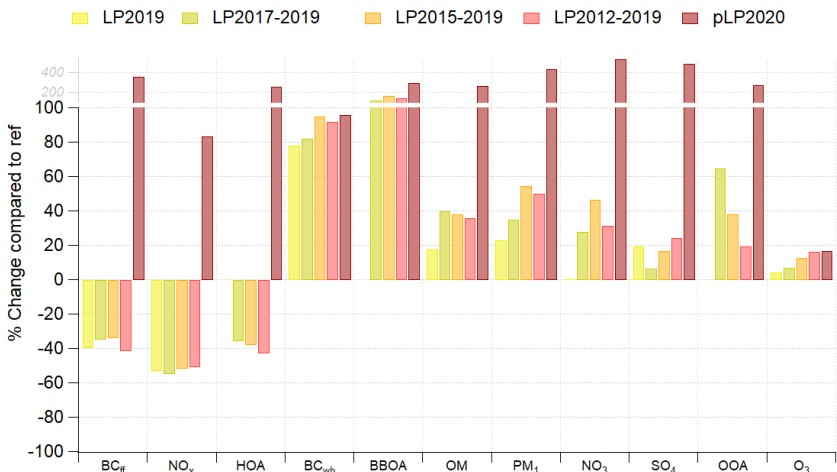

**Figure 1 : Relative concentration change (%) of each specie used in this study following different reference**
**periods.**

Alternatively, we tried to use the weeks corresponding to LP for previous years (e.g. 2017-2019, LP2017-2019) as the reference. Obtained results seemed to reveal a substantial decrease for the concentrations of pollutants related to traffic emissions (i.e., $BC_{ff}$, HOA and $NO_x$), but clear increases of all other investigated

pollutants, especially secondaries (Fig. 1). Similar results were obtained when narrowing (e.g., looking only at $LP_{2019}$, Mesas-Carrascosa et al., 2020) or broadening (e.g., $LP_{2012-2019}$ or $LP_{2015-2019}$, Adams, 2020; Sun et al., 2020) the reference period.

Applying such methodologies blindly encompasses interannual variability, and do not take potential long-term trends into account. Moreover, they inherently assume comparable meteorology. Figure 2a presents

the trajectory density during LP2020, showing a strong prevalence of continental air masses. When comparing the air mass origin of LP2020 and other reference periods (pLP2020, LP2019, LP2017-2019, LP2015-2019, LP2012-2019, Figure S3), it appears that the continental sector is under-represented, while the oceanic sector is over-represented (Fig. 2b). The unrealistic features of pLP2020 can then be explained by a drastic change of Western Europe meteorological conditions (from low-pressure to high-pressure system)

concomitantly with the application of lockdown policy measures in France, preventing for using such a





methodology in the present study. For the other reference periods, it may also explain why these methodologies are associated to an increase of eg $NO_3$, $SO_4$ and OOA, due to an underestimation of business-as-usual concentrations for LP2020 meteorological conditions. For this matter, and in order to avoid erroneous, contradictory and/or counterintuitive results, it appears critical to take synoptic circulation and air mass origin into account.

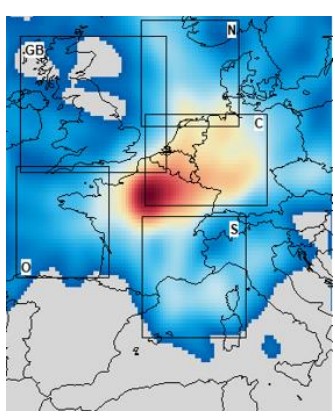

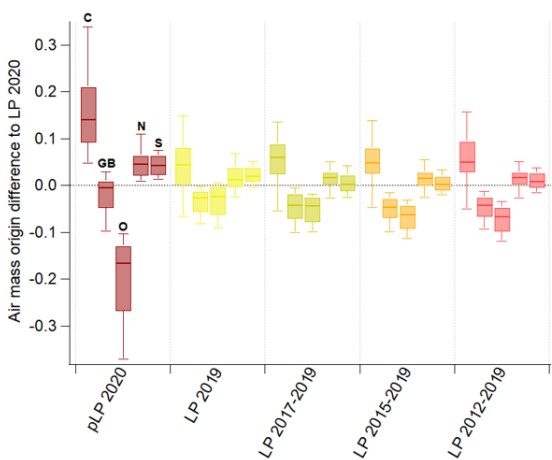

**Figure 2: a) Air mass origin during LP2020, represented as trajectory density. b) Difference of air mass origin between LP2020 and the different reference periods (trajectory densities have been normalized to unity), divided into 5 main sectors (Continental, Great Britain, Oceanic, North, and South) delimited in a).**

Additionally, April 2020 in France was exceptionally warmer (+4.5°C), drier (-43% of precipitation in the Paris region) and more sunny (+60%) than usual (1981-2010 climatological reference values). To this perspective, local meteorology needs to be taken into account as much as possible, since this climatological extreme may have favoured photochemical processes.

To overcome all these issues and account for the strong synergy between PM chemical composition, emission sources and meteorology, we developed the Analog Application for Air Quality (A³Q) method, which is described below.

### 3.2. The "Analog Application for Air Quality" (A³Q) approach

#### 3.2.1. Description

Analogs of atmospheric circulation (Yiou et al., 2013; Lorenz, 1969; Van Den Dool, 1994; Zorita and Storch, 1999; Cattiaux et al., 2012) have been widely used for different climatological purposes, notably for atmospheric reconstructions and in the characterization of the role of synoptic circulation in extreme meteorological events (Yiou et al., 2013; Vautard et al., 2018). Circulation analogs are generally computed from daily pressure spatial distributions. Here, we built the analogy based on three successive layers:





*Synoptic*. Circulation analogs are computed similarly to previous studies. We used daily sea-level pressure (SLP) data, to better characterize near-surface atmospheric circulation as our study covers near-surface pollutants. The SLP data is extracted from NCEP/NCAR reanalysis data (Kalnay et al., 1996) along the

historical period that covers 2012 to 2019. The SLP fields considered here have a horizontal resolution of 2.5 x 2.5° and cover a spatial domain ranging -20°W to +15°E in longitude and +40°N to +60°N in latitude. This region is chosen because it includes atmospheric pressure patterns that influence near-surface wind (Raynaud et al., 2017) in our area of study. The study period covers 92 days from March 1st until May 31st 2020, while the historical period covers the same months in 2012-2019. Fifty best circulation analogs (minimum

spatial correlation of 0.5) are sought for each day of the study period, using the spatial correlation as a way to measure similarity between SLP fields. The calendar distance between the day in the study period and days in the historic period is maximum 30 days. Out of 50 potential days with analog atmospheric circulation, only those with a spatial correlation higher than 0.6 (representing 97.8% of all analog days, 23.2 analogs/day on average) were kept.

*Regional*. Air mass trajectory (AMT) of each synoptic analog was compared to the AMT of the corresponding lockdown day. For each day, trajectory density (log of the occurrence of trajectory endpoints) was calculated over a 0.5°x0.5° grid covering Western-Europe. The distribution of spatial correlation values between each day of LP and each analog day is presented in Fig. S4. Since this distribution is rather spread out, a low threshold at 0.2 (representing 70% of analog days, 16.3 analogs/day on average) was selected in order to

remove the worst analog AMT, but also to keep sufficient variability. An example of satisfactory and unsatisfactory analogs is shown in Fig. S5.

*Local*. A specific constrain on local ambient temperature and Relative Humidity (RH) was implemented. Indeed, both variables are key drivers of the partitioning of semi-volatile material, such as ammonium nitrate. Therefore, a satisfactory representation of local meteorological conditions by the analogs is needed in order

to robustly capture and characterize any change of concentration. Over the study period, the analog performance regarding temperature and RH respectively ranges from -15°C to +15°C, and from -36% to +64%. Concretely, analogs that are much colder and wetter than the observation day may be associated to enhanced condensation of semi-volatile compounds, which would lead to an overestimation of the estimated decrease of e.g. nitrate. This specifically occurred on April 21st, 22nd and 23rd 2020. To avoid that issue, we excluded

from the list the analogs having the 5% worst performance. Acceptable ranges were therefore ]-9.3, 6[ and ]-19, 35[ for temperature and RH, respectively. Despite these relatively wide ranges, the A³Q methodology allows to efficiently reconstruct meteorological conditions during the lockdown period. Indeed, Mean Bias of -0.17 m/s, -1.52°C, -1.3 hPa and 8.7% were respectively obtained for wind speed, ambient temperature, pressure and relative humidity, indicating a satisfactory analogy. Sensitivity tests presented below also

demonstrate that stricter ranges do not significantly change the analog results.

### 3.2.2. Data preparation

Daily averages have been computed for each variable. A valid average is considered if at least 75% of the day is covered.





Because the dataset consists in multi-year observations, long-term trends may impact the estimated concentration change due to lockdown. To that end, a seasonal Mann-Kendall test was performed on each variable on the 01/2012-02/2020 period. The Mann-Kendall R package was used (Collaud Coen et al., 2020), which includes three pre-whitening approaches in order to reduce the weight of autocorrelation. Results are summarized in Table 1. From this analysis, $NO_x$, OA, HOA, OOA, $O_3$, $BC_{ff}$ concentrations (2012-2020, including

lockdown) were linearly corrected on a daily basis.

| Variable | Sen's slope ($\mu g/m^3$/year) | MK p-value (95% confidence interval) |
|---|---|---|
| $NO_x$ | –0.156 | 0.009 |
| $NO_3$ | –0.037 | 0.22 |
| $SO_4$ | – | – |
| OA | –0.068 | <0.001 |
| HOA | –0.019 | 0.049 |
| BBOA | –0.004 | 0.18 |
| OOA | –0.068 | <0.001 |
| $O_3$ | 0.459 | 0.01 |
| $PM_1$ | – | – |
| $BC_{wb}$ | –0.002 | 0.067 |
| $BC_{ff}$ | –0.018 | <0.001 |

**Table 1. Sen's slope (in $\mu g/m^3$/year) and MK p-value for each variable of the study. No slope means that the result doesn't reach the 95% confidence interval. Slope is statistically significant for p < 0.05.**

In order to limit the unwanted weight of positive outliers which have poor statistically representativity, the 1% highest daily concentrations of each variable were removed.

### 3.2.4. Sensitivity tests

The results presented here primarily depend on the list of analog days that is calculated. The overall analog

number is at first determined by the strictness of the correlation coefficients of atmospheric circulation and air mass origin. Selection of best analogy leads to poor statistical representativeness (Fig. S6), with a low number of analog days. It is instead preferable to remove analog days associated with worst trajectory correlation (Fig. S4). It is noteworthy that little change in the correlation coefficients (Table 2, Scenario 1-4) has little impact on the results (for all variables) presented in this manuscript (Fig. S7). This can be mainly

related to the reasonable change in the number of analog days.

Similarly, the impact of subsequent filtering with temperature and relative humidity were also investigated. To this end, two additionnal scenarios were considered (S5-6, Table 2). S5 and S6 have limited impact on traffic-related variables (Fig. S7), on the contrary to wood-burning tracers and secondary compounds. This especially highlights the essential role of meteorological representativeness in order to characterize the

changes of secondary pollution. Indeed, when no temperature and RH filtering is performed (Scenario 5), highest decrease of $NO_3$ is linked to analog days that are associated to higher RH (Fig. S8a) and lower temperature (Fig. S8b), which favor the partitioning of nitrate in the particulate phase. Scenario 6 has a stricter filtering than the Base one, and exhibits very good performance regarding the reconstruction of





meteorological conditions (Table 3). However, we show that both scenarios have very similar daily $NO_3$

concentration change despite slight discrepancies in meteorological performance. This underlines that the

thresholds used in the Base scenario are sufficient to provide robust results.

| | Synoptic analog R | Trajectory analog R | RH acceptability range | T acceptability range | Min. Analog nb |
|---|---|---|---|---|---|
| Base | 0.6 | 0.2 | ]-19, 35[ | ]-9.3, 6[ | 5 |
| Scenario 1 | 0.6 | 0.3 | ]-19, 35[ | ]-9.3, 6[ | 5 |
| Scenario 2 | 0.6 | 0.1 | ]-19, 35[ | ]-9.3, 6[ | 5 |
| Scenario 3 | 0.5 | 0.2 | ]-19, 35[ | ]-9.3, 6[ | 5 |
| Scenario 4 | 0.7 | 0.2 | ]-19, 35[ | ]-9.3, 6[ | 5 |
| Scenario 5 | 0.6 | 0.2 | - | - | 5 |
| Scenario 6 | 0.6 | 0.2 | ]-19, 17[ | ]-8, 6[ | 3 |

**Table 2. Acceptability thresholds used in different scenarios to evaluate the sensitivity of our methodology.**

| | RH Mean Biais (%) | Temperature Mean Biais (°C) | Wind Speed Mean Biais (m/s) | Pressure Mean Biais (hPa) |
|---|---|---|---|---|
| Base | -8.7 | +1.52 | -0.17 | -1.3 |
| Scenario 5 | -10.5 | +2.1 | -0.14 | -0.52 |
| Scenario 6 | -1.95 | +1.2 | -0.21 | -2.35 |

**Table 3. Performance (expressed as Mean Biais) of different scenarios to predict meteorological parameters during
lockdown**

### 3.2.5. Performance evaluation

Furthermore, the performance of the analog methodology has been evaluated on a business-as-usual period,

from 01/01/2020 to 01/03/2020. The construction of the analog list went through the same steps, with the

same thresholds. The acceptability range for T and RH moved to ]-5, 4[ and ]-16, 19[, respectively. This can

be primarily related to the less extreme climatological conditions of Jan.-Feb. 2020. Mean Bias (MB),

Normalised Mean Bias (NMB), Pearson correlation coefficient (r), and the fraction of data within a ratio of 2

(FAC2) have been used for the evaluation.

Scatter plots and metric values are respectively presented in Figure 3 and Table 4. Results indicate

satisfactory performance of all variables during the evaluation period, although the range of observed

concentrations remains rather low.

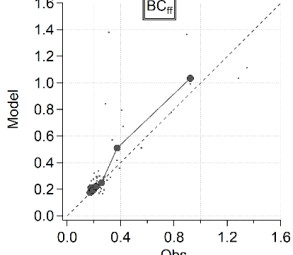 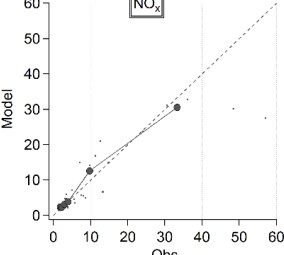 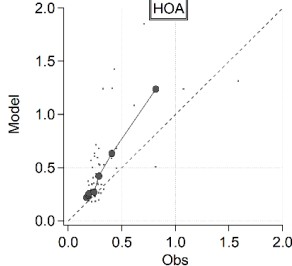





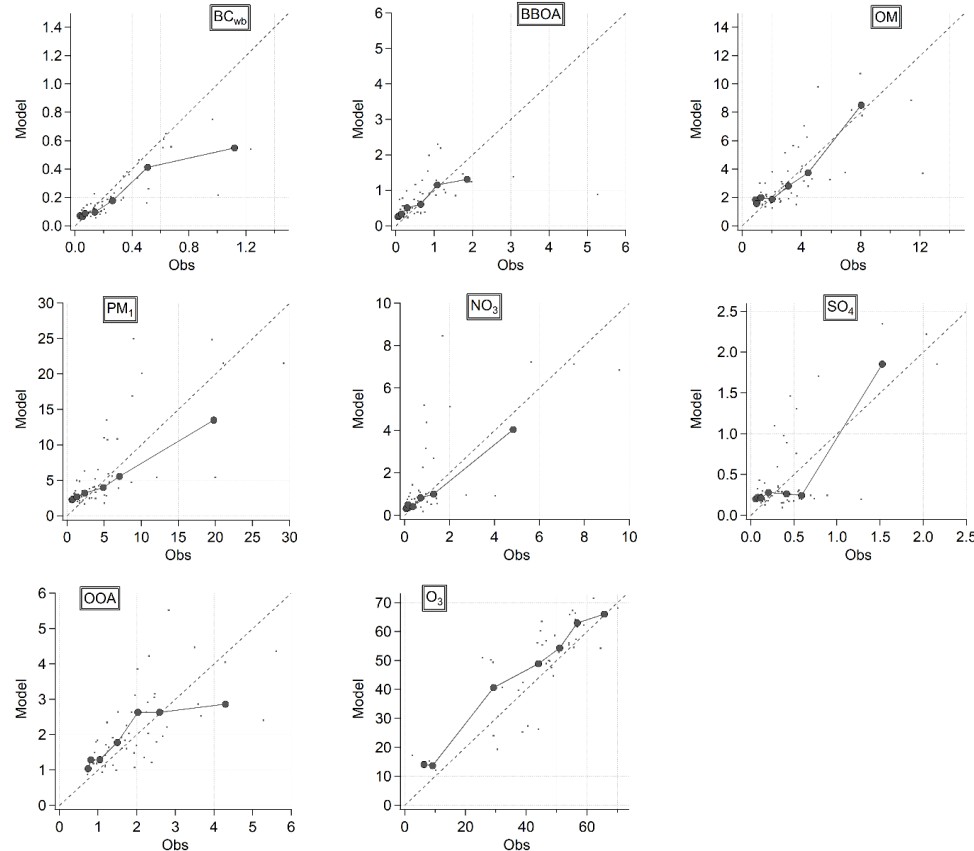

**Figure 3. Scatter plots of observed versus estimated concentrations by A³Q during the evaluation period of Jan.-Feb. 2020.**

|  | MB (μg/m³) | NMB (%) | FAC2 | R |
|---|---|---|---|---|
| BCff | 0.049 | 15.3 | 0.96 | 0.78 |
| NOx | −0.53 | −7.1 | 0.95 | 0.90 |
| HOA | 0.17 | 53.0 | 0.79 | 0.65 |
| BCwb | −0.06 | −24.1 | 0.82 | 0.82 |
| BBOA | 0.04 | 6.0 | 0.60 | 0.45 |
| OM | 0.03 | 0.8 | 0.93 | 0.69 |
| PM₁ | 0.89 | 17.2 | 0.68 | 0.73 |
| NO₃ | 0.38 | 36.1 | 0.54 | 0.71 |
| SO₄ | 0.03 | 7.9 | 0.46 | 0.70 |
| OOA | 0.19 | 9.8 | 0.98 | 0.63 |
| O₃ | 4.27 | 9.7 | 0.95 | 0.84 |

**Table 4. Metric values (MB, NMB, FAC2 and r) for the evaluation of A³Q during Jan.-Feb. 2020.**





## 4. Results and discussion


Figure 4 presents the distribution of absolute concentration change for each specie during lockdown. Results are further discussed in the following subsections.

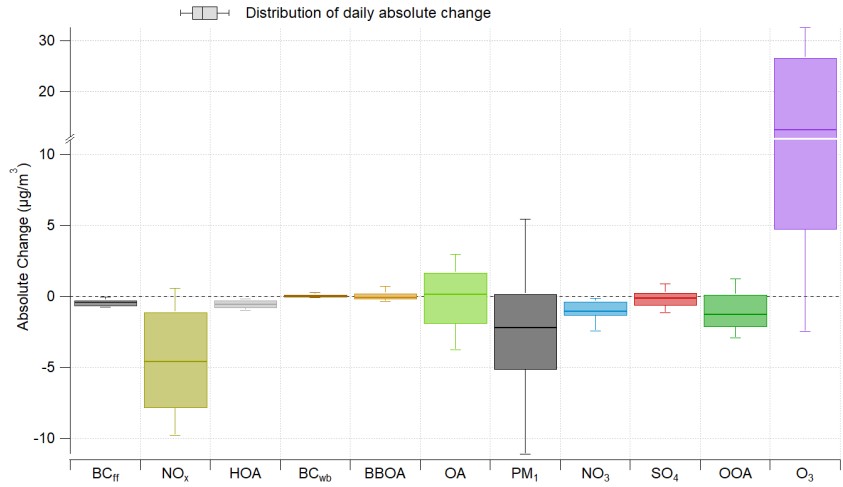

**Figure 4: Absolute changes of ambient concentrations of reactive gases and particulate pollutants due to lockdown. Boxplots represent the distribution of daily absolute change (µg/m³); 10th, 25th, 50th, 75th and 90th percentiles were used. Values are presented in Table 5.**

| | Absolute change (µg/m³) | | | | | Median relative change (%) | p-value |
|---|---|---|---|---|---|---|---|
| | p10 | p25 | p50 | p75 | p90 | | |
| BCff | -0.74 | -0.65 | -0.42 | -0.28 | -0.06 | -54.8 | $3.2 \times 10^{-15}$ |
| NOx | -9.79 | -7.81 | -4.57 | -1.1 | 0.58 | -42.7 | $2.14 \times 10^{-5}$ |
| HOA | -0.98 | -0.80 | -0.52 | -0.27 | -0.17 | -61.8 | $1.2 \times 10^{-13}$ |
| BCwb | -0.11 | -0.03 | 0.04 | 0.13 | 0.24 | 20.1 | 0.003 |
| BBOA | -0.36 | -0.20 | -0.03 | 0.22 | 0.72 | 12.0 | 0.44 |
| OA | -3.77 | -1.89 | 0.16 | 1.66 | 2.95 | -6.6 | 0.71 |
| PM1 | -11.10 | -5.13 | -2.16 | 0.17 | 5.45 | -14.9 | 0.02 |
| NO3 | -2.42 | -1.33 | -1.03 | -0.37 | -0.16 | -45.5 | 0.0067 |
| SO4 | -1.14 | -0.61 | -0.11 | 0.28 | 0.90 | -10.3 | 0.21 |
| OOA | -2.92 | -2.12 | -1.26 | 0.12 | 1.21 | -25.1 | 0.002 |
| O3 | -2.49 | 4.74 | 12.57 | 26.70 | 32.62 | 20.2 | $4.45 \times 10^{-9}$ |

**Table 5. Distribution of absolute concentration change (10th, 25th, 50th, 75th, 90th percentile has been used), as well as**
**the median relative change (%). p-values (95% confidence interval) of the pairing Student t-test between Observations and Analog timeseries during lockdown**





### 4.1. Primary sources

Species usually considered as markers for primary traffic emissions ($NO_x$, $BC_{ff}$ and HOA) exhibit at
SIRTA a median decrease of concentrations by 42-62% (Table 5) at SIRTA. For $NO_x$, this is very consistent
with previous results in the Paris region using machine learning approaches (-42%, Grange et al., 2020).
Moreover, the $NO_x$ decrease at SIRTA, a peri-urban background station, also matches the relative decrease
calculated for urban and peri-urban stations across the region (-42% and -39%, respectively), although
absolute changes are graduated (-11 µg/m$^3$ and -16.4 µg/m$^3$ for urban and peri-urban stations, respectively).
Although the intensity of the decrease has differed from site to site, the temporality of the change was
uniform at the regional scale, including SIRTA (Fig. 5). It is also consistent with traffic counting data in the
Paris region (https://dataviz.cerema.fr/trafic-routier/), with a slow traffic increase throughout the
lockdown period.

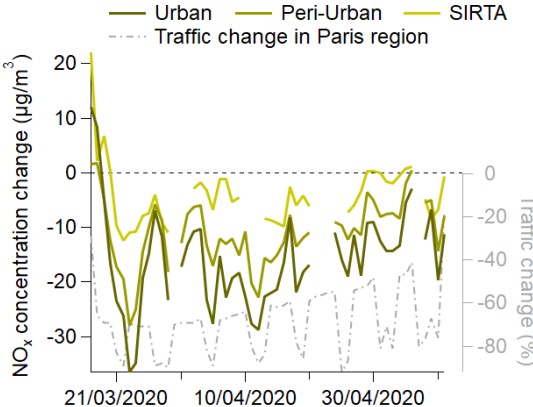


**Figure 5: Temporal variation of $NO_x$ concentration changes at SIRTA, and urban and periurban background stations of the Paris region.**

On the other hand, wood burning tracers ($BC_{wb}$ and BBOA) exhibit an increase of +20% and +58%
respectively, which can be primarily related to the stay-at-home order, enhancing emissions of residential
heating (Grange et al., 2020). Although absolute change is limited (Fig.4), by converting $BC_{wb}$ to $PM_{wb}$, and $BC_{ff}$
to $PM_{ff}$, increased $PM_{wb}$ concentrations compensated or even exceeded the decrease of $PM_{ff}$ during specific
days (Fig. 6). At the same time, the mean weekly variation of wood burning changed during lockdown (Fig.
S9), with increased concentrations during the week, compared to the relatively flat variation in business-as-
usual conditions (e.g. +67% on Fridays). Therefore, lockdown changed both intensity and temporality of the
wood burning source in the Paris region.





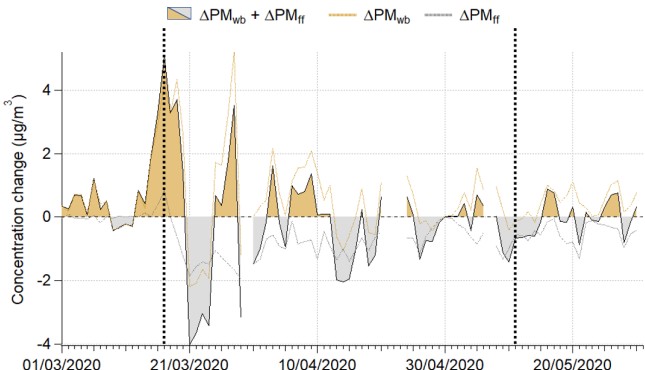

**Figure 6 Temporal variations of $\Delta PM_{wb}$ + $\Delta PM_{ff}$ during lockdown. Brown shaded area shows compensation of wood burning. Individual $PM_{wb}$ and $PM_{ff}$ concentration change are also displayed with dotted lines. Black vertical lines**
**delimit the start and the end of the lockdown period.**

### 4.2. NOₓ-induced influence on secondary pollutants

#### 4.2.1. *Ozone*

Nitrogen oxides play a central role within the atmospheric reactor, enabling the formation of secondary
pollutants (Kroll et al., 2020) such as tropospheric ozone and secondary organic and inorganic aerosols (SOA
and SIA, respectively). Ozone is found to increase by 20% (Fig. 4). This negative feedback, due to the titration
effect of NO, is already well characterized (Reis et al., 2000), and the magnitude of the change inversely
follows well the one of $NO_x$ (Fig. 7a). A sharp-enough decrease of $NO_x$ shall tip over ozone formation to a $NO_x$-
limited system (Markakis et al., 2014), which may be seen from this relationship, where ozone concentration
change stabilizes at +20 µg/m³ for $\Delta NO_x$ below -10 µg/m³ at SIRTA. Nevertheless, it is worth highlighting
that the climatological extreme of Spring 2020, with strong positive temperature anomalies, should have also
contributed to increased $O_3$ concentrations, as previously emphasized for Europe (Meleux et al., 2007). Even
though meteorology is taken into account by the $A^3Q$ method, the unique character of this springtime
heatwave inherently blurs the discrimination between ambient temperature and $NO_x$ impacts on ozone
formation. This limitation would also occur for any other statistical approaches, such as machine learning.
Figure 7b shows indeed that some of the highest $\Delta O_3$ values are associated to high temperatures (daily
average > 18°C), concomitantly with substantial decrease of $NO_x$ concentrations. On the other hand, positive
$\Delta O_3$ are also obtained for rather low temperature (< 10°C) and highest decreases of $NO_x$, which means that in
this case the increase of $O_3$ shall be only $NO_x$-related.


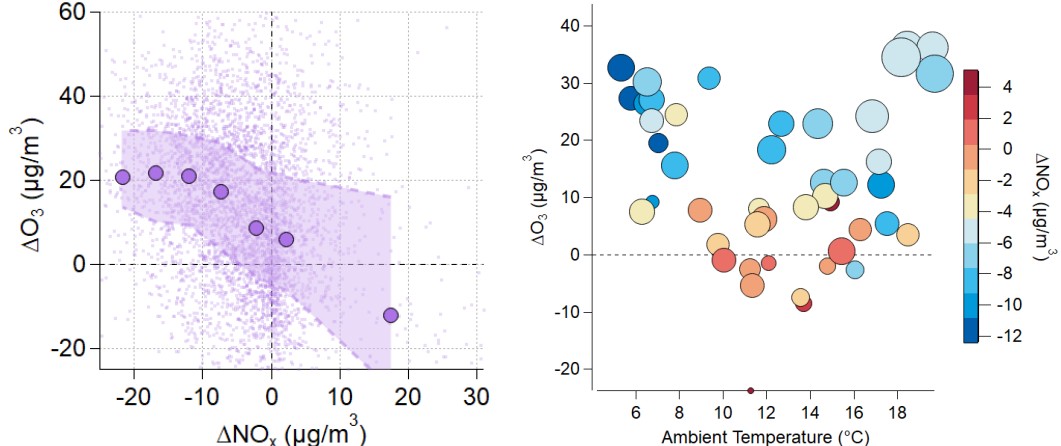

**Figure 7 a) Concentration change (µg/m³) of O₃ versus concentration change of NOₓ (µg/m³). Coloured dots correspond to a 100 resampling following the inverse normal distribution law, whose mid-height width is the standard deviation of each analog day. Markers represent the median, bottom and top and shaded area the 25th and 75th percentile, respectively. b) Concentration change of O₃ versus ambient temperature. Marker's color and size are respectively function of ΔNOₓ and observed O₃ concentrations.**

### 4.2.2. Particulate Nitrate

Springtime in the Paris region is usually associated with PM pollution episodes that are mainly triggered by particulate ammonium nitrate (Beekmann et al., 2015) ($NH_4NO_3$), resulting from the reaction between $HNO_3$ ($NO_x$ oxidation) and ammonia ($NH_3$). For that matter, agricultural activities (the major source of $NH_3$ in Western-Europe (Fortems-Cheiney et al., 2016)) were neither stopped nor restrained during lockdown. Therefore, business-as-usual ammonia concentrations can reasonably be assumed, and since the formation regime of nitrate in Paris has previously been found to be $NO_x$-limited (Petetin et al., 2016), a change of regime to $NH_3$-limited is highly unlikely (Viatte et al., 2021). On median, nitrate decreases by 45%. The decrease linearly follows the one of $NO_x$ (Fig. 8), although both compounds differ in terms of reactivity and footprint. A similar relationship is shown when using urban and peri-urban $NO_x$ concentrations (Fig. S10), which is consistent with the temporal correlation of $\Delta NO_x$ throughout the Paris region (Fig. 5). However, a slight shouldering of the decrease for urban $NO_x$ can be noticed, which implies that this efficiency regarding $NO_3$ is related to the amplitude of $NO_x$ reduction.

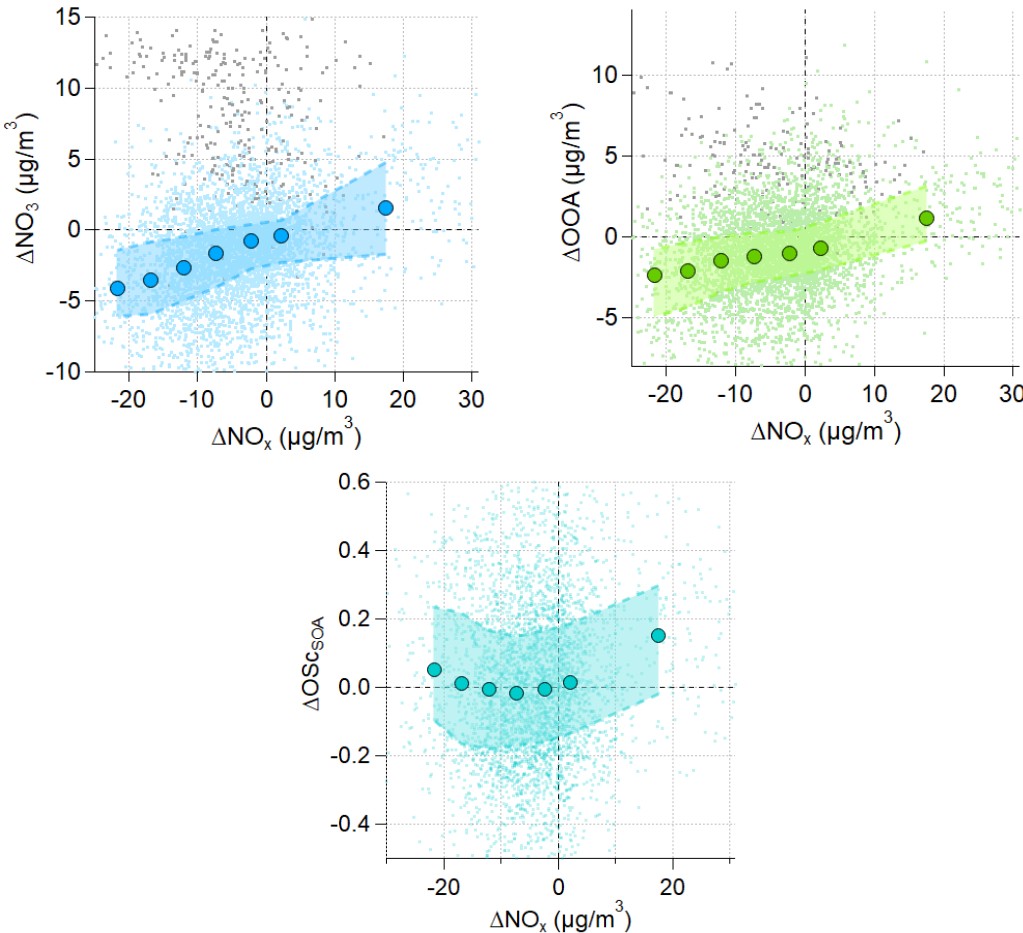

**Figure 8** Same as Fig.8a, for NO₃, OOA and OSc_SOA. Grey dots correspond to days with predominant long-range transport, identified by a positive NO₃ and OOA concentration change concomitantly with a SO₄ concentration peak (Fig. 10), which have been excluded from percentile calculations. Markers represent the median, bottom and top and shaded area the 25th and 75th percentile, respectively.

This result suggests the importance of rapid ammonium nitrate formation at a rather local scale (Petit et al., 2015; Wang et al., 2020), but shall not eclipse continental advection that can occur in Paris, especially during Springtime (e.g. Beekmann et al., 2015). To that end, sulphate shows little change (-8%) with no clear statistical significance ($p > 0.05$), which indicates a similar influence of long-range pollution advection, and that $SO_2$ sources in Eastern Europe (Pay et al., 2012) may not have experienced a significant decrease. Consistent information on that matter are unfortunately still rather scarce at the European scale. Filonchyk et al. (2020) recently showed an increase of $SO_2$ in several Polish urban areas, although their methodology does not take meteorology into account. The Carbon Monitor initiative (Liu et al., 2020b) also records a decrease of 11.5% of the Power sector in Europe during 2020 compared to 2019. On specific days, positive peaks of $\Delta NO_3$ are concomitant to higher $SO_4$ concentrations (Fig. 9), which means that nitrate was in these cases mainly advected from long-range transport, despite a decrease of $NO_x$. It is also concomitant with a higher Nitrogen Oxidation Ratio (NOR=$NO_3/(NO_2+NO_3)$) values, and highest positive change (top panel of Fig. 9). Given the





$NO_x/NO_3$ relationship (Fig. 8a), and hypothesizing that a decrease of locally-formed $NO_3$ is always associated
to a decrease of $NO_x$ concentration at the measurement site, long-range transported $NO_3$ can be assumed to
overcompensate the regional decrease. For instance, on March 28[th] and April 19[th], the total $\Delta NO_3$ respectively
of 11.7 and 6.7 µg/m³ could be apportioned into a regional decrease of -5.5 and -3.7 µg/m³, with an advected
contribution of 17.2 and 10.4 µg/m³, respectively.


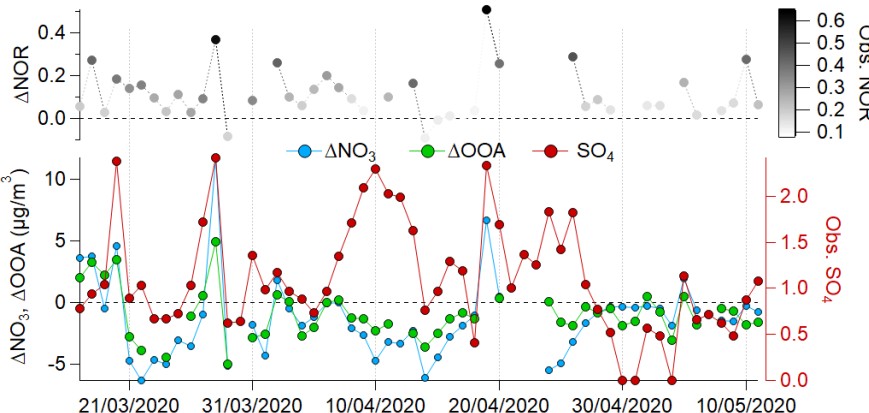

**Figure 9 Temporal variation during LP2020 of (bottom) $\Delta NO_3$, $\Delta OOA$ and $SO_4$ concentrations (µg/m³); (top) $\Delta NOR$**
**values, color-coded by observed NOR.**

4.2.3. *Secondary Organic Aerosols*

Secondary organic aerosols, proxied by OOA concentrations, exhibit a decrease of 25% compared to
business-as-usual conditions. Given the multitude of SOA precursors and formation pathways, this decrease
can be linked to numerous factors. Indeed, Srivastava et al. (2019) recently highlighted the complexity of the
SOA fractions in the Paris region during Springtime. The lack of specific organic tracers prevents from
thoroughly apportioning SOA over the long-term analysis period of 2012-2020. As a consequence, the
apparent decrease of 25% may derive from several compensatory feedbacks, which cannot be individually
characterized here. Although SOA in Paris can't be only related to traffic (Crippa et al., 2013; Srivastava et al.,
2019), the decrease of OOA seems to be also correlated with $\Delta NO_x$ (Figure 8b). $NO_x$ steps in SOA formation
notably when reacting with peroxy radicals (R-$O_2$), resulting from VOCs oxidation with the hydroxyl radical.
The exceptional amount of sunshine during lockdown may have positively influenced the availability of OH[•]
for the initialization of SOA formation. However, no direct observations available at SIRTA can support this
assumption directly. The odd oxygen $O_x$ (= $O_3$ + $NO_2$), a conservative tracer of photochemical chemistry (Sun
et al., 2020), shows a slight increase (+9%). But unlike the findings of Herndon et al. (2008) in Mexico-city
during Spring, only a moderate correlation is found with OOA ($r^2$=0.4, slope = 0.068, following the





recommendation of removing wood-burning and long range transport related episodes). This may be linked to the rather low $O_x$ concentrations (60-110μg/m$^3$) during the lockdown period.

425         Despite very limited change on average (<1%), daily $OSc_{SOA}$ is found to differ from business as usual conditions, adopting a comma-like shape, function of $\Delta NO_x$ (Fig. 8c). Indeed, although $\Delta OSc_{SOA}$ decreases until $\Delta NO_x > -7$ μg/m$^3$, it increases back for higher $NO_x$ decreases, reaching a median of +0.05. Despite being moderate, this behaviour may reflect some changes in SOA chemistry, notably regarding gas-phase oxidation of anthropogenic precursors, as highlighted in Herndon et al. (2008). But at this stage, the whole equation

comprises too many unknown variables (from VOC precursors to particulate end products) in order to fully describe the impacts of lockdown measures onto the numourous SOA formation mechanismsremains arduously apprehendable.

**4.3. Implications for air quality**

The lockdown enforced during Spring 2020 in Paris corresponds to a real-life emission scenario, representing the extreme case of a quasi-total interruption of the vehicular traffic source. Up to now, no mitigation policy could have gone that far. From the results presented here, the impact of $NO_x$ control on submicron primary traffic-related aerosols ($BC_{ff}$ and HOA) remains limited, mainly because both don't

contribute much to $PM_1$ concentrations during Spring (5.0 and 5.7% respectively). However, it is worth mentioning that a higher impact can be expected within $PM_{10}$ in the vicinity of traffic, related to the decrease of non-exhaust emissions (brake/tire wear and resuspension). This fraction has recently been highlighted as particularly harmful for human health (Daellenbach et al., 2020); therefore $NO_x$ control shall provide a significant co-benefit for that specific matter.

Reducing the concentrations of secondary compounds is an arduous task, because mitigation policies can inherently only focus on the reduction of primary pollutants. But regional $NO_x$ reduction appears to have the potential to be an efficient mitigation policy regarding secondary aerosols (mainly $NO_3$ and SOA to a lesser extent), which account on average for more than half of $PM_1$ during Spring at SIRTA (respectively 28.9% and 31.4%). Moreover, although the traffic ban has been applied consistently over the lockdown period, the

decrease of $NO_x$ concentrations at SIRTA ranges from 0 to around -20 μg/m$^3$. Adding the impact of $NO_x$ on $BC_{ff}$, HOA, $NO_3$, $NH_4$ (considering neutralized aerosols) and OOA together, the corresponding change of PM-related material ranges from -2 to -9 μg/m$^3$ (Fig. 10). The efficiency of $NO_x$ mitigation shall therefore be put in perspective with meteorological conditions (e.g. horizontal dispersion, since lowest $\Delta NO_x$ values are associated to continental air masses), as well as the vertical atmospheric dynamic (Dupont et al., 2016). It is

also worth mentioning that in the case of long-range pollution transport episodes, substantial efforts to reduce emissions at the city scale will not be enough to counterbalance additional advected material, as shown previously for $NO_3$ and OOA.



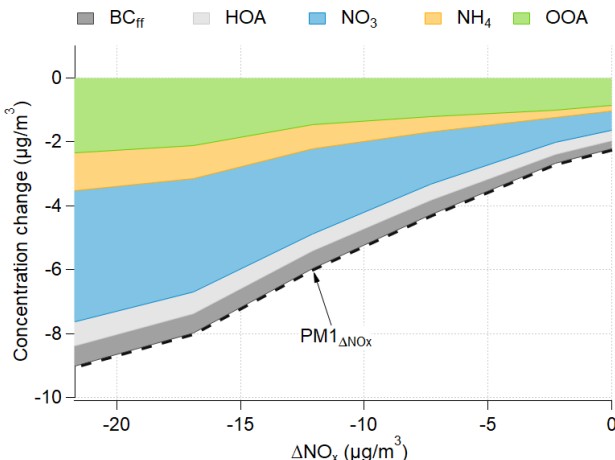

**Figure 10 Impact of regional NOₓ concentration change on PM-related species (BCff, HOA, NO₃, NH₄, OOA). ΔNH₄ was estimated from ΔNO₃, assuming aerosol neutralization (Petit et al., 2015).**

### 5. Conclusion

The COVID-19 pandemic led to life-changing restrictions, with strict stay-at-home orders in Western-Europe during Spring 2020. As a consequence, light-duty vehicular traffic was almost completely stopped in urban areas, such as the Paris region. Despite tragic death records, lockdowns represent an open-air mitigation experiment, in a period that is usually associated with PM pollution episodes dominated by secondary material. However, the characterization of a change in chemical composition is not

straightforward because meteorology can strongly contribute to the temporal variability of atmospheric pollutants. To that end, a unique methodology was built in order to compare each day of lockdown with analog days, having similar meteorology. The analogy was based on three successive meteorological layers, where synoptic, regional and local meteorology was considered. This innovative approach was applied to a comprehensive in-situ dataset, acquired at the SIRTA observatory, a sub-urban station located 20km South-

West of Paris. The A³Q method provided very satisfactory results over a business-as-usual period, which ensures a robust characterization of concentration changes in the Paris region during lockdown.

    The unprecedented drop of traffic commuting due to the stay-at-home order led to an expected decrease of primary traffic pollutants, NOₓ (-42%), HOA (-62%) and BCff (-55%), as well as an increase of ozone concentrations due to the lesser contribution of the O3 sink by NO. Concomitantly, primary particles related

to residential wood-burning shows a slight increase (+12-20%). The decrease of NOₓ (-42%) triggered positive feedbacks regarding secondary aerosols especially nitrate and SOA, which are found to drop by 42% and 25% respectively. A NOₓ relationship suggests the significance of rather local pollution formation, contrasting with previous results in the region. The decrease of NO₃ was compensated sporadically during long-range transport episodes. Oxidation state of SOA is also seen to vary with NOₓ concentration change, but our

understanding of the involved phenomena still remains limited, notably due to the lack of long-term VOC in-situ observations in the Paris region.



Finally, the proposed A³Q method may be considered as an efficient tool to monitor and quantify more precisely the impact of lockdowns in other urban areas. It could also be useful for the quantitative evaluation

of emergency mitigation policies settled during pollution episodes.


*Author contributions.* J.-E.P., O.F, V.G., Y.Z., J.S., L.S., F.T., N.B., T.A. and J.-C.D. contributed to the availability of in-situ measurements at SIRTA. Y.Z. performed the source apportionment analysis between 2012 and 2020. R.V. provided the

list of analog days from synoptic circulation. J.-C.D. and M.H. demonstrated the feasibility of the analog method on SIRTA in-situ data. J.-E.P. performed the additional analyses, with contributions from L.S. J.-E.P. wrote the paper with the assistance from all authors.

*Competing interests.* The authors declare no competing interests.


*Data availability.* In-situ measurements at SIRTA are available through the EBAS database (https://ebas.nilu.no). Ozone data from Airparif are available on https://www.airparif-asso.fr
GDAS files for backtrajectory calculation are available on https://www.arl.noaa.gov/hysplit/hysplit/
ZeFir procedure is available on https://sites.google.com/site/zefirproject/


*Acknowledgments.* The authors would like to thank Robin Aujay-Plouzeau, Roland Sarda-Estève, Dominique Baisnée and Vincent Crenn for their contribution in maintaining data acquisition at SIRTA. Christophe Boitel and Marc-Antoine Drouin are acknowledged for their support in data management. This work also greatly benefited from discussions within

the COLOSSAL COST action CA16109.

*Fundings.* This research has been supported by the EUFP7 and H2020 ACTRIS projects (grant nos. 262254 and 654109), by the National Center for Scientific Research (CNRS), by the French alternatives energies and Atomic Energy Commission (CEA), by the French Ministry of Environment, and by the DIM-R2DS program from the Ile-de-France region.

The authors gratefully acknowledge CNRS-INSU for supporting measurements performed at the SI-SIRTA, and those within the long-term monitoring aerosol program SNO-CLAP, both of which are components of the ACTRIS French Research Instructure, and whose data is hosted at the AERIS data center (https://www.aeris-data.fr/).




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
