# Peer review of "Response of atmospheric composition to COVID-19 lockdown measures during Spring in the Paris region (France)"

_Atmospheric Chemistry and Physics, 2021_

## Author Comment (AC2)

**We sincerely thank Referee#2 for the improvement suggestions. Below is a detailed answer to each comment which was raised. Changed made to the manuscript are reported in red hereafter.**

Major comments:

The impacts of lockdown on aerosol chemistry were evaluated using Analog Application for Air Quality (A3Q) approach. Please give a detail discussion on the reliability and uncertainty (or limitation) of the method. I suppose that the method of comparing with reference period could be moved to the supplementary material because the focus should be on A3Q.

**We added a further discussion on A3Q in the conclusion, as follows:**

"The A³Q method provided satisfactory results over a business-as-usual period, which ensures a robust characterization of concentration changes in the Paris region during lockdown. Yet, A³Q requires a 9+ long term dataset, otherwise results can rapidly suffer from shortfall of representativeness. Also the analogy needs to be carefully inspected, notably in terms of local meteorology. Indeed, the first synoptic layer appears to be not quite enough to capture all the specificity of the sampling site."

**Given the comments of Reviewer #1, we would like to keep in the main text some discussion about reference periods. But we tried to be a bit more concise.**

Please elaborate how to calculate the absolute and relative changes of aerosol and gaseous species due to lockdown.

**We added the equations in the main text for clarity.**

The Aerosol Chemical Speciation Monitor (ACSM) were used to measure aerosol composition. Since there are two versions of ACSMs nowadays, i.e., quadruple and Time-of-Flight ACSM, the author should clarify this in the Method. A composition dependent CE or a constant CE is used in this study?

**We clarified this in the text**

For Sec. 2.2, the POA-constrained PMF (or ME-2) was performed on OA matrix to resolve three OA factors (i.e., HOA, BBOA, OOA) during January-May 2020. The source apportionment results during June 2011- March 2018 were obtained from previous studies. It

is unclear whether similar PMF method and OA components were used between the two periods. Please specify it.

**Indeed, the harmonization of PMF analysis between both studies was not described. Both used the a-value approach (Canonaco et al., 2015), with 2 constrained factors, HOA and BBOA. Same Reference spectra were used, and the used a-values were in the same order of magnitude. As a result, the ouput profiles are very consistent, with slope and regression coefficients higher than 0.9 (see below). We added these pieces of information in the text accordingly :**

"Results obtained here enrich the existing timeseries (Zhang et al., 2019) from June 2011 to March 2018.(where MO-OOA and LO-OOA were summed as OOA). Both PMF outputs were obtained with the same reference profiles of HOA and BBOA (Fröhlich et al., 2015), and similar a values for HOA and BBOA were used (on average 0.26 and 0.32 for HOA and BBOA, respectively; 0.21 and 0.22 in Zhang et al., 2019). As a result, HOA and BBOA profiles are very consistent, with slope and r² higher than 0.9."

The elemental ratio and oxidation degree of SOA were calculated using I-A method. Which fragments or m/z were used for O:C and H/C calculation? Considering that ACSM detect species with unit mass resolution, I am afraid that the absolute values of element ratios were questionable to some extent.

**For OSc$_{SOA}$, we used the m/z presented in Canagaratna et al. (2015), i.e. m/z 29, 43 and 44. Given the unit mass resolution of the ACSM, the calculation has higher uncertainty, and the absolute values are indeed questionable. That is why we don't discuss absolute values, but only variations of values, as currently stated in the manuscript. We made it clearer in the text:**

"From there, O:C$_{SOA}$, H:C$_{SOA}$ and OSc$_{SOA}$ were calculated from the Improved Aiken (Aiken et al., 2008) equations provided in Canagaratna et al. (2014), using m/z 29, 43 and 44. Given the unit mass resolution of the instrument, it is important to underline that these equations provide only qualitative information for ACSM data. Absolute values, most probably associated with significant uncertainties will therefore not be discussed here. Nevertheless, it is sufficient to characterize a change, since they are uniformly applied throughout the dataset."

Figure 3, the scatter plots of simulated versus observed species contain two markers, i.e., small dots and six solid dots. Please explain it in the figure caption.

**We clarified the caption**

Compared to the secondary inorganic aerosol, both POA factors and OOA showed much lower R during the evaluation period (Table 4). Did this mean that the organics might not be well reproduced by this model? If this, the quantification of OA changes during 2020 lockdown might also be affected.

**For the evaluation period, HOA and OOA have a R value of 0.65 and 0.63, respectively. It is indeed lower than the value of 0.90 found for $NO_x$, but in the same order of magnitude for OM (0.69), $NO_3$ (0.71) or $SO_4$ (0.70). BBOA has however a lower correlation coefficient (0.45).**

**The lower performances of OA factors may be related to the absence of 2019 data. Then, fewer analog days may lead to higher dispersion and uncertainty. However, they are associated with satisfactory Mean Bias, Normalized MB and FAC2 values. Moreover, HOA and BBOA are consistent with other tracers (eg BCff and BCwb respectively).**

**That is why we believe that the quantification of OA changes during 2020 lockdown is robust.**

**We added some lines on BBOA in the text as follows:**

"The lower performance of BBOA in terms of co-variations may be related to the absence of 2019 data, where fewer analog days could lead to higher dispersion, but also to the fickleness of the wood-burning source. Still, as presented in the Results section, BBOA variations are consistent with $BC_{wb}$."

The time periods used for calculation and discussion were a bit confused. In line 205-210, the author said that "the study period covers 92 days from March 1st –May 31st 2020". However, January-February 2020 was chosen for model performance evaluation. Why only January-February 2020 was used? Which period was referred to as "lockdown" through the study? Please declare it in the Method.

**Indeed, that sentence in line 205-210 was confusing. We removed it.**

**The lockdown period (LP2020) is already defined in the introduction. The Reviewer questions our 2-month evaluation period, being too short, but it is not clear why it would be so. In Grange et al. (2020), the evaluation period was 15 days (February 14[th] to March 1[st]). Petetin et al. (2020) used the 2.5 monthes before lockdown to evaluate their approach. This is reminded in the text as follows:**

"Furthermore, the performance of the analog methodology has been evaluated on a business-as-usual period, from January 1st, 2020 to March 1st, 2020, similarly to the work of Petetin et al. (2020) and Grange et al. (2020)"

**It is not clear which evaluation period would be sufficient for the Reviewer.**

In line 390-400, is there other evidence that NO3 formed from long-range transportation?

**SO$_4$ concentrations in Paris region have already been proven to be mainly related to long range transport (eg Favez et al., 2021). The cluster analysis presented in Figure S3b also supports this finding. We emphasized this in the text:**

"On specific days, positive peaks of $\Delta NO_3$ are concomitant to higher SO$_4$ concentrations (Fig. 9). Since SO$_4$ has been previously found to be mainly advected in Northern France (eg Favez et al., 2021), also supported by the cluster analysis in Fig. S3b, this means that nitrate was in these cases mainly advected from long-range transport, despite a decrease of NO$_x$."

The total changes of NO3 can be quantitatively apportioned into regional decrease and advected contribution? Please elaborate the estimation method.

**Yes, this is part of the assumption. We detailed a bit more the estimation as follows:**

Given the NO$_x$/NO$_3$ relationship (Fig. 8a), and hypothesizing that a decrease of locally-formed NO$_3$ is always associated to a decrease of NO$_x$ concentration at the measurement site, long-range transported NO$_3$ can be assumed to overcompensate the regional decrease (eq. 4).

$$\Delta NO_3^{total} = \Delta NO_3^{advected} + \Delta NO_3^{local}$$ equation 4

Where $\Delta NO_3^{total}$ is the daily concentration change at t$_i$, calculated from A$^3$Q. $\Delta NO_3^{local}$ is calculated from the relationship with $\Delta NO_x$ (Figure 8).

For instance, on March 28$^{th}$ and April 19$^{th}$, the total $\Delta NO_3$ respectively of 11.7 and 6.7 µg/m$^3$ could be apportioned into a regional decrease of -5.5 and -3.7 µg/m$^3$, with an advected contribution of 17.2 and 10.4 µg/m$^3$, respectively. This result would need to be further investigated and confirmed from eg Chemistry Transport Model simulations, but still underlines the deleterious impact of long-range transport.

Minor comments:

Please define BCff and BCwb when first mentioned.

**Now defined when first mentioned.**

The table caption should be placed on the top to table instead of the bottom.

**We changed this in the text.**

Although this is not a final publication version of ACP, the author should carefully check the output styles of the references as ACP recommended. For example, complete and abbreviated journal names

**The bibliography has been re-build using harmonized information.**

The abstract could be more simplified, particularly for the descriptions before A3Q method.

**The beginning of the abstract now reads as follows:**

"Since early 2020, the COVID-19 pandemic has led to lockdowns at national scales. These lockdowns resulted in large cuts of atmospheric pollutant emissions, notably related to the vehicular traffic source, especially during Spring 2020. As a result, air quality changed in manners that are still currently under investigation. The robust quantitative assessment of the impact of lockdown measures on ambient concentrations is however hindered by weather variability. In order to circumvent this difficulty, an innovative methodology has been developed. The Analog Application for Air Quality (A3Q) method is based on the comparison of each day of lockdown to a group of analog days having similar meteorological conditions. "

---

## Author Comment (AC3)

**We sincerely thank Referee#1 for the improvement suggestions. Below is a detailed answer to each comment which was raised. Changed made to the manuscript are reported in red hereafter.**

Scientific Merit:

Significant changes in meteorology would be expected in the transition from winter to spring (i.e., pre-lockdown period and lockdown period) and even from early to late spring (i.e., beginning of lockdown period to end of lockdown period). Thus, the pre-lockdown and lockdown comparisons in pollutant concentrations (Fig. 1) and air mass trajectories (Fig. 2) are important for illustrating the point, but should not be a main focus for justifying the A3Q approach. For example, in lines 171-173 it is stated that the continental sector is under-represented and the oceanic sector over-represented when comparing the lockdown period with other reference periods. This is clear for the pre-lockdown period; however, within the measurement uncertainty there appears to be no statistically significant difference for the continental sector across the other reference years (over the same date range) and statistically different but (potentially) small differences for the oceanic sector in three of the four periods.

**The strong differences with pLP2020 actually flatten the differences with other reference periods. Still, we agree that this analysis may not be quantitative enough to prove the misrepresentation of the continental sector, although no measurement uncertainty is plotted in Figure 2 because no measurements were used in the trajectory analysis. We replaced it by a trajectory cluster analysis over LP2012-2020. The occurrence of continental air masses during LP2020 is around 28%, while it is 13-18% for the reference periods of previous years. The oceanic sector is inversely over-represented. The cluster analysis also reveals that highest $NO_3$ concentrations are associated to continental air masses (median of 6.6 µg/m$^3$), which contrasts with the median value of 0.6 µg/m$^3$ for the oceanic sector. Therefore, an under-representation of the continental sector, as well as an over-representation of the oceanic sector, should lead to an underestimation of business as usual $NO_3$ concentrations during LP2020.**

**Also following the comment of Reviewer 2, we changed section 3.1 as follows:**

The assessment of lockdown impact on air quality lies on the use of a reference period, which is assumed to be representative of business-as-usual conditions during LP2020, following:

$$\%change = 100 \cdot \frac{LP_{2020} - ref}{ref} \qquad \text{equation 1}$$

In the current literature, different "reference periods" are used, from a "pre-lockdown" period (pLP2020, Toscano and Murena, 2020; Dantas et al., 2020; Otmani et al., 2020), to the weeks corresponding to LP of previous years (e.g. 17/03 to 11/05 during 2017-2019 is LP2017-2019). Nevertheless, in the case of SIRTA, applying these methodologies unquestioningly, without verifying the inherent

hypothesis that data are comparable, can lead to significant variability, and counterintuitive results. Figure 1 presents concentration relative changes for the SIRTA dataset, using pLP2020, LP2019, LP2017-2019, LP2015-2019 and LP2012-2019 as references. Significant increases for all pollutants (e.g., + 83% in $NO_x$, +439% in $PM_1$) are found with pLP2020, which seems to contradict the observed drop of traffic. For the other reference periods, results reveal a substantial decrease for the concentrations of pollutants related to traffic emissions (i.e., $BC_{ff}$, HOA and $NO_x$), but clear increases of all other investigated pollutants, especially secondaries.

[Figure]

Figure 1 : Relative concentration change (%) of each specie used in this study following different reference periods, as well as from the A3Q approach presented in this article

As reference periods, they assume meteorological conditions representive of LP2020. However, April 2020 in France was exceptionally warmer (+4.5°C), drier (-43% of precipitation in the Paris region) and sunnier (i.e. hours of sunshine during the day; +60%) than usual (1981-2010 climatological reference values). Table S1 presents the meteorological variability of the different reference periods (in terms of ambient temperature, RH, pressure and wind speed), and shows that they don't reproduce the meteorology of LP2020 (in terms of min, max and average, especially T and RH), and also fail at reproducing its temporality (low r values). Moreover, from a trajectory cluster analysis (Fig. S3a), it appears that they misrepresent the variability of air mass origin. The unrealistic features of pLP2020 can indeed be explained by a drastic change of Western Europe meteorological conditions (from low-pressure to high-pressure system) concomitantly with the application of lockdown policy measures in France (Fig. 2). For the other reference periods, they still under-represent the continental sector (13-18%) compared to

LP2020 (28%), and inversely over-represent oceanic air masses. Given the fact that, for instance, $NO_3$, $SO_4$ and OOA exhibit highest concentrations with continental air masses, these methodologies at SIRTA can most likely underestimate business-as-usual concentrations, and therefore lead to erroneous results.

[Figure]

Figure 2. Frequency of trajectory clusters for each LP periods.

To overcome all these issues and account for the strong synergy between PM chemical composition, emission sources and meteorology, we developed the Analog Application for Air Quality ($A^3Q$) method, which is described below.

Similar comparisons for temperature and precipitation for the LP and references periods would be valuable.

**A similar comparison for ambient temperature, RH, wind speed and pressure also highlights that these reference periods shall not be used in the case of SIRTA. We mentioned this in the text (see previous comment), and added a table in the supplementary:**

| | | LP2020 | LP2019 | LP2017-2019 | LP2015-2019 | LP2012-2019 |
|---|---|---|---|---|---|---|
| Temperature (°C) | min | 4.2 | 5.31 | -0.5 | -0.5 | -0.5 |
| | max | 19.7 | 19.8 | 23.2 | 23.2 | 23.2 |
| | mean | 12.9 | 10.6 | 11.1 | 11.0 | 10.6 |
| | MB | | 2.2 | 1.8 | 1.9 | 2.3 |
| | r | | 0.17 | 0.4 | 0.45 | 0.45 |

| | | | | | | |
|---|---|---|---|---|---|---|
| RH (%) | min | 34.0 | 36.9 | 36.9 | 36.9 | 34.9 |
| | max | 87.5 | 84.7 | 92.0 | 96.6 | 100 |
| | mean | 56.0 | 65.2 | 66.0 | 67.1 | 69.0 |
| | MB | | -9.1 | -9.9 | -11.0 | -13.1 |
| | r | | 0.36 | 0.32 | 0.28 | 0.30 |
| Wind Speed (m/s) | min | 0.8 | 0.3 | 0.3 | 0.3 | 0.3 |
| | max | 6.9 | 5.2 | 5.2 | 7.5 | 7.5 |
| | mean | 2.5 | 2.5 | 2.5 | 2.6 | 2.6 |
| | MB | | -0.03 | -0.01 | -0.13 | -0.16 |
| | r | | -0.09 | 0.07 | 0.31 | 0.23 |
| Pressure (hPa) | min | 982.0 | 975.4 | 975.4 | 975.4 | 971.5 |
| | max | 1009.0 | 1012.5 | 1014.5 | 1014.7 | 1015.5 |
| | mean | 996.5 | 996.0 | 995.7 | 996.0 | 995.1 |
| | MB | | 0.9 | 0.72 | 0.49 | 1.43 |
| | r | | 0.13 | 0.03 | 0.08 | 0.05 |

Table S1: Meteorological conditions during LP2020, LP2019, LP2017-2019, 2015-2019 and 2012-2019. Min, max and average values for Temperature, RH, Wind speed and Pressure are presented for each period. MB and r are calculated through through a daily reconstruction of daily values of LP2020.

Minor comment-the date range listed in 161 does not match any of those in Fig. 1.

**As stated above, we changed section 3.1, but tried to make the date range clearer**

"[…] to the weeks corresponding to LP of previous years (e.g. 17/03 to 11/05 during 2017-2019 is LP2017-2019) "

The authors need to better demonstrate that the A3Q approach provides a significantly better solution than comparing with a range or ranges of previous years over the same days (i.e., see Parker et al. GRL, 2020).

**From the changes presented above, we think that it is now clear that meteorology needs to be taken into account, especially for secondary pollutants, whose concentrations at the receptor site highly depends on meteorological conditions. To this respect, lockdown periods of previous years can't be considered as representative of business-as-usual periods for the case of SIRTA. Then, we agree that the question "what is the performance of A3Q regarding meteorological parameters" would deserve a more detailed answer in the text. We changed this accordingly, with an additional figure in the supplementary:**

"Despite these relatively wide ranges, the $A^3Q$ methodology allows to efficiently reconstruct meteorological conditions during the lockdown period. Indeed, Figure S11 presents, for the Jan.-May 2020 period, the temporal variations of observed and estimated Temperature, RH and Pressure. It shows low Mean Bias values, as well as satisfactory correlation coefficients (r value of 0.78, 0.82 and 0.63, respectively), which indicates a satisfactory analogy. Sensitivity tests presented

below also demonstrate that stricter ranges do not significantly change the analog results. "

[Figure]

Figure S11. Temporal variations of ambient temperature, RH and pressure during January–May 2020, observed (black) and estimated by A³Q (blue)

It would be useful to see some of the pollutant concentrations presented in Fig. 1 with the results using the different analogs (Fig. 7).

**Figure 1 has been changed to add the results from A³Q.**

presentation quality:

Clarity of the manuscript could be improved with organizational changes and additional editing. The language in many places is quite challenging, and the main points are obscured or unclear.

The introduction, as written, includes one paragraph about COVID-19 restrictions, perturbations in human activity, and associated opportunities to better understand atmospheric composition and chemistry. Indeed, there have been a large number of studies published on air quality during COVID-19, some of which are cited by the authors. It is suggested that the introduction be reorganized such that these two pieces (motivation and current published studies) be combined as a single paragraph.The introduction could then be expanded to describe the current state-of-the-science regarding air quality in France/Europe and the importance of

considering meteorology in air quality studies, which are both relevant to the conclusions and only cursorily described in the second paragraph of the introduction. A more concise discussion of secondary chemistry and PM composition would also strengthen the introduction and the manuscript as a whole.

**Thank you for the suggestions. The introduction now reads as follows:**

"With the worldwide spreading of the SARS-COV-2 coronavirus, the COVID-19 outbreak has been responsible of millions of premature deaths. In order to slow down contagion rates, social interactions have progressively been limited until the establishment of strict lockdowns at national scales (Anderson et al., 2020) enforced during several weeks, especially during Spring 2020 in Europe. The corresponding stay-at-home orders resulted in a sudden halt of economic activities, and, as a consequence, in an unprecedented drop of emission of pollution sources. To this perspective, and despite tragic death records, these lockdowns are unique opportunities to characterize an extreme end of mitigation policy scenarios, and future low-carbon megacities from direct observations. Scientific initiatives are thriving across the globe in order to assess the impact of lockdowns on air quality. They report, for most, a sharp decrease of nitrogen oxides (NOx) concentrations, as well as an increase of troposheric ozone (e.g. China: Le et al. (2020); India: Mahato et al. (2020); USA: Liu et al. (2020a); Europe: Sicard et al. (2020); Grange et al. (2020); South-America: Siciliano et al. (2020)) as a response to stay-at-home orders.

The increase of ozone is one counterintuitive example of the complex chemistry occurring within the atmosphere, although its link with the decrease of NOx concentrations has been well established (eg Reis et al., 2000). As highlighted by Kroll et al. (2020), beyond NOx, O3 and PMx, additional information are needed in order to further characterize the impacts of lockdown on the atmospheric chemical system. Indeed, PM is composed of several different fractions, from organic to inorganic, and from primary to secondary pollutants, with diverse sources and transformation processes. Any concentration change of PM may derive from various compensatory feedbacks which are not characterized, limiting therefore our understanding of the impacts of lockdown on air quality. Moreover, Springtime in North-Western Europe is usually associated with high PM pollution episodes dominated by secondary material (mainly ammonium nitrate and sulfate, and secondary organic aerosols -SOA) as shown in Bressi et al. (2021). Ammonium nitrate is formed in the atmosphere from the neutralization of nitric acid (formed through NOx oxidation) with ammonia. The comprehensive characterization of SOA formation is also blurred by the overwhelming numbers of transformation pathways, precursors as well as oxidant availability. Thus far, only few studies have investigated the impacts of lockdown on PM chemistry and sources in Asia (eg Chang et al., 2020; Sun et al., 2020; Tian et al., 2021; Manchanda et al., 2021) by comparing the lockdown period with other periods (either a pre-lockdown period, or the same period of the year of previous years).

On the other end, the assessment of air quality implications of large cuts in urban pollutant emissions is strongly hampered by meteorological variability, which is one of the main drivers of air pollution temporality. For instance, unfavourable meteorology has previously been associated to increase of PM concentrations in various urban areas worldwide (eg Dupont et al., 2016; Wang et al., 2020; other ref). Sun et al. (2020) also highlighted severe hazes during lockdown in China, linked to stagnant meteorological conditions. Therefore, without climatologically representative values, specific care must be considered when comparing concentrations observed during and outside the lockdown period. The robustness of this assessment depends on the way meteorology is handled and on what "reference period" is chosen to compare with the "lockdown period". A recent review by Gkatzelis et al. (2020) pointed out that, despite the luxuriance of scientific literature, more than half of examined articles didn't take meteorology into account. Advances on machine-learning (ML) approaches have however enabled to disentangle the contributions of meteorological conditions on the temporal variations of primary and secondary PM components (e.g. Stirnberg et al., 2021). ML has successfully been applied mainly on NOx and O3 in various European urban areas (Petetin et al., 2020; Grange et al., 2020). But weather-corrected studies of PM chemistry are still scarce, especially in Western-Europe.

The present study aims at reconciling a robust and innovative methodology with a quasi-comprehensive in-situ dataset, acquired within the Paris region (France). The 12-million inhabitants of the region, representing around 20% of the total French population, were placed under lockdown from March 17th, 2020 to May 10th, 2020, further designated as LP2020."

There are several statements that are unclear as written. A few examples follow, but this is not an exhaustive list. It is recommended that the manuscript be reviewed for these unclear or ambiguous statements.

Lines 49-51: "Without climatologically representative values, comparisons of concentrations observed during and outside the lockdown periods shall thus free themselves from differences in weather." The authors are making the valid point that care must be taken when comparing air quality measurements over different time periods because much of the variability can be driven by meteorology. However, this specific sentence seems to contradict that point, though it is very difficult to interpret.

**We rephrased the sentence as follows:**

"Therefore, without climatologically representative values, specific care must be considered when comparing concentrations observed during and outside the lockdown period"

Lines 55-58: The authors state- "Air quality shall not be restrained to NOx, O3, and PMx only, and limited number of studies so far has treated air quality as a whole, notably taking PM chemistry into account." The sentences before and after suggest that the authors are referencing the need to consider speciation of PM. Depending on the objectives of a study, this may be a critical aspect, but isn't necessarily a requirement for air quality studies.

**We agree. We rephrased the sentence as follows:**

"As highlighted by Kroll et al. (2020), beyond $NO_x$, $O_3$ and $PM_x$, additional information are needed in order to further characterize the impacts of lockdown on the complex atmospheric chemical system."

Lines 176-177: "…it may also explain why these methodologies are associated to an increase of eg NO3, SO4 and OOA, due to an underestimation of business-as-usual concentrations for LP2020 meteorological conditions." It seems the authors here are suggesting that if the air masses for LP2020 were correctly represented/compared, and if there was no lockdown, then the NO3, SO4, and OAA concentrations in the reference periods may not have been higher than business-as-usual LP2020. However, there was no business-as-usual LP2020. Maybe the authors mean to suggest that the differences in concentrations, without accounting for meteorology/air mass origin, are not only due to changes in human activity and are likely exaggerated due to fewer air masses of continental origin in prior years.

**Yes, we definitely agree. Section 3.1 has been rewritten, provifing further justifications.**

minor comments:

lines 79-83: The latter part of this sentence is unclear-it is clear that the measured fractions were corrected for collection and ionization efficiency, but it is not clear what it meant by "successfully participated in" intercomparison. Is this for the measurements being reported or is this something that has been done previously?

**We rephrased the sentence as follows:**

"and showed satisfactory performances during ACTRIS intercomparaison exercises (Crenn et al., 2015; Freney et al., 2019)."

line 140: Please provide some additional details regarding ZeFir.

**We added additional details about ZeFir, as follows:**

"Calculations using HYSPLIT executables were automatically controlled by ZeFir (Petit et al., 2017a), a user-friendly interface based on Igor Pro 6.3 (Wavemetrics©). The cluster analysis presented in section 3.1 was also applied from HYSPLIT executables, controlled by ZeFir. Five clusters were used (Fig. S3a), in accordance with the Total Spatial Variance (TSV). The two oceanic cluster were summed as one."

line 186: How is "sunny" quantified?

**It is quantified as the number of hours of sunshine during the day. We made it clearer in the text.**

"Additionally, April 2020 in France was exceptionally warmer (+4.5°C), drier (-43% of precipitation in the Paris region) and sunnier (i.e. hours of sunshine during the day; +60%) than usual (1981-2010 climatological reference values)"

line 227: Why would wetter days lead to enhanced condensation of semi-volatile compounds?

**Higher ammonium nitrate concentrations could be expected on days with higher RH, due to its hygroscopicity. Higher concentrations could also occur on colder days due to its semi-volatile properties. We corrected this in the text.**

"analogs that are much colder and wetter (higher RH) than the observation day may be associated to enhanced condensation of semi-volatile and/or hygroscopic compounds, which would lead to an overestimation of the estimated decrease of e.g. nitrate "

lines 230-231: Annotation (brackets) is unclear (also in Table 2).

**They mean "excluded". We made it clearer in the text.**

"Acceptable ranges were therefore between the 5th and 95th percentiles (excluded) of ΔT and ΔRH values, which were respectively ]-9.3, 6[ and ]-19, 35["

section 3.2.4-The presentation of the sensitivity tests is confusing as written, with both "S" used to indicate a scenario and also "scenario".

**We replaced "S" with "scenario" throughout the section for consistency.**

line 295: "specie" should be "species"

**Changed.**